

# Shallow water heterobranch sea slugs (Gastropoda: Heterobranchia) from the Región de Atacama, northern Chile

Juan Francisco Araya[1,2] and Ángel Valdés[3]

[1] Departamento de Geología, Universidad de Atacama, Copiapó, Región de Atacama, Chile
[2] Programa de Doctorado en Sistemática y Biodiversidad, Universidad de Concepción, Concepción, Chile
[3] Department of Biological Sciences, California State Polytechnic University, Pomona, California, United States

## ABSTRACT

The coast of northern Chile has been sparsely studied in regards to its invertebrate fauna, with just a few works reviewing the distribution of local mollusks. This work presents a survey of the shallow water heterobranch sea slugs currently occurring around the port of Caldera (27 °S), in the Región de Atacama, northern Chile. Eight species of sea slugs were found in this study: *Aplysiopsis* cf. *brattstroemi* (*Marcus, 1959*), *Baptodoris peruviana* (d'Orbigny, 1837), *Diaulula variolata* (d'Orbigny, 1837), *Doris fontainii* d'Orbigny, 1837, *Onchidella marginata* (Couthouy in *Gould, 1852*), *Phidiana lottini* (*Lesson, 1831*), *Tyrinna delicata* (*Abraham, 1877*) and the new species *Berthella schroedli* sp. nov., described herein. All of the species found in the area are endemic to South America, having distributions in the southeastern Pacific and South Atlantic Oceans, from Ancash, Perú to Peninsula Valdés, Argentina, and two of them represent species which are endemic to the Chilean coasts (*Aplysiopsis* cf. *brattstroemi* and *Berthella schroedli*). The finding of a previously undescribed species emphasizes the need of further surveys, particularly in subtidal and deeper waters, in order to improve the knowledge on this neglected fauna in Atacama.

## INTRODUCTION

The mollusks of the Región de Atacama, in northern Chile, have been sparsely studied; most of the species commonly present in the area were described in the nineteenth century (*Broderip & Sowerby, 1832*; *Sowerby, 1832*; *Sowerby, 1833*; *d'Orbigny, 1834–1847*; *Gould, 1852*; *Hupé, 1854*; *Gay, 1854*, *Philippi, 1860*, among others), with a few works reviewing species during the past century (*Dall, 1909*; *Gigoux, 1932*; *Gigoux, 1934*; *Rehder, 1945*) and, more recently, with several works describing new species (*Osorio, 2012*; *Araya, 2013*; *Araya, 2015a*; *Araya, 2015b*; *Miquel & Araya, 2013*; *Collado, 2015*; *Araya & Reid, 2016*) or giving new records (*Araya & Araya, 2015a*). Regarding heterobranch sea slugs in particular (sensu *Camacho-García et al. (2014)* and *Padula, Wirtz & Schrödl (2014)*),

Corresponding author
Juan Francisco Araya,
jfaraya@u.uchile.cl

**Table 1 Heterobranch sea slugs found in the Region of Atacama, northern Chile; species, distribution, ecology and references.** Occurring species involve those cited by *Marcus (1959)*, *Schrödl (1996a)*, *Schrödl (2003)*, and material examined in this work.

| Species | Distribution | Ecology | References |
|---|---|---|---|
| *Aplysiopsis cf. brattstroemi* (*Marcus, 1959*) | Antofagasta (23°39′S; 70°25′W) to Bahia de Coliumo (36°32′S; 72°57′W), Chile | Sea floor, subtidal | *Schrödl (1996a)* |
| *Baptodoris peruviana* (d'Orbigny, 1837) | San Lorenzo (12 °S), Peru to Valparaiso, Chile (33°02′S; 71°38′W) | Sea floor, epifaunal, subtidal | *Fischer & Cervera (2005a)* and *Fischer & Cervera (2005b)* |
| *Berthella schroedli* sp. n. | Caldera (27 °S), Chile | Under sunken rocks, infaunal, subtidal | This work |
| *Diaulula variolata* (d'Orbigny, 1837) | Ica (14 °S), Perú to Bahía de San Vicente (36 °S), Chile | Sea floor, epifaunal, subdtidal | *Fischer & Cervera (2005a)*, *Fischer & Cervera (2005b)* and *Uribe et al. (2013)* |
| *Doris fontainii* (d'Orbigny, 1837) | Islote Ferrol (09°08′22″S; 78°37′15″W), Ancash, Peru to northern Argentina | Sea floor, epifaunal, subtidal | *Uribe et al. (2013)* and *Valdés & Muniáin (2002)* |
| *Onchidella marginata* (Couthoy in *Gould, 1852*) | Iquique (20 °S), Chile to Isla de los Estados (coordinates), Argentina | Under rocks, epifaunal, intertidal | *Rosenfeld & Aldea (2010)* |
| *Phidiana lottini* (*Lesson, 1831*) | Callao (12°02′S), Peru to Comau Fjord (42°15′S; 72°25′12″W), Chile | Sea floor, epifaunal, subtidal | *Schrödl et al. (2005)*, *Uribe et al. (2013)* and *Schrödl & Hooker (2014)* |
| *Tyrinna delicata* (*Abraham, 1877*) | Isla Blanca (09 °S), Ancash, Peru to Peninsula Valdés, in the Atlantic Magellan Strait | Sea floor, epifaunal, subtidal | *Schrödl & Millen (2001)* and *Uribe et al. (2013)* |

only the studies by *Bergh (1898)*, *Marcus (1959)*, *Schrödl (1996a)*, *Schrödl (1996b)*, *Schrödl (1997)*, *Schrödl (2003)*, *Fischer, van de Velde & Roubos (2007)* and most recently *Labrín, Guzmán & Sielfeld (2015)* have included species from northern Chile. However, a few recent papers dealing with the Peruvian fauna, including some species commonly found in Chilean waters (e.g., *Millen et al., 1994*; *Nakamura, 2006*; *Nakamura, 2007*; *Martynov & Schrödl, 2011*; *Uribe & Pacheco, 2012*; *Uribe et al., 2013*; *Schrödl & Hooker, 2014* and others) have also contributed to the knowledge of this group in the southeastern Pacific.

The present study provides records of sea slugs found in shallow waters around Caldera (27 °S), Región de Atacama, northern Chile. The coast of this area consists of rocky formations with sparse sandy beaches and a comparatively narrow intertidal zone. Rocky platforms, boulder fields and intertidal pools are common; however, some sheltered areas have open sandy beaches, usually exposed to strong surf. All of the species reviewed in this work are endemic to southern South America; with two of them presenting new distributional records in Chile (Table 1). The aim of this preliminary study is to contribute to the knowledge of the molluscan fauna in Chile, particularly from the largely neglected northern coasts.

## MATERIALS AND METHODS

The material examined was collected in the summers of 2010–2012 in diverse locations near the port of Caldera (27 °S), Region of Atacama, northern Chile. All the collecting was made manually in the intertidal areas, mostly on rocky outcrops and tidal pools. The specimens were deposited in the collections of the Museo de Paleontología de Caldera (MPCCL), Caldera, Chile; Museo de Zoología de la Universidad de Concepción (MZUC),

Concepción, Chile, California State Polytechnic University Invertebrate Collection (CPIC), Pomona, USA, and in the collection of the Natural History Museum of Los Angeles County Museum (LACM), Los Angeles, USA. Field study permits were not required for this study and none of the species studied herein are currently under legal protection. All the collected specimens were preserved in 95% ethanol. Photographs of living animals were taken with a Canon A530 digital camera and a 10× geologic loupe. All sizes given are living measurements, radular features were examined by scanning electron microscopy (SEM). Color plates were composed with basic image programs and colors of the images were not modified.

In order to characterize genetically and barcode the new species of *Berthella*, DNA extraction was performed using a hot Chelex® protocol. Approximately 1–3 mg tissue was taken from one animal and cut into fine pieces for extraction, the tissue was rinsed and rehydrated using 1.0 mL TE buffer (10 mM Tris, 1 mM EDTA, pH 8.0) for 20 min. A 10% (w/v) Chelex® 100 (100–200 mesh, sodium form; Bio-Rad) solution was prepared using TE buffer. After rehydration, the mixture was then centrifuged, 975.00 mL of the supernatant was removed, and 175.00 mL of the Chelex® solution was added. Samples were then incubated at 56 °C in a water bath for 20 min, heated to 100 °C in a heating block for 8 min, and the supernatant was used for PCR. Folmer's universal COI primers (*Folmer et al., 1994*) were used to amplify the region of interest for one specimen. The master mix (for each sample) was prepared using 34.75 μL $H_2O$, 5.00 μL PCR Buffer (ExACTGene; Fisher Scientific), 5.00 μL 25 mM $MgCl_2$, 1.00 μL 40 mM dNTPs, 1.00 μL 10 μM primer 1, 1.00 μL primer 2, 0.25 μL 5 mg/mL Taq, and 2.00 μL extracted DNA. Reaction conditions were an initial denaturation for 3 min at 95 °C, 39 cycles of 1) denaturation for 45 sec at 94 °C, 2) annealing for 45 sec at 45 °C, and 3) elongation for 2 min at 72 °C, and a final elongation for 10 min at 72 °C. PCR products yielding bands of appropriate size (approximately 695 bp) were purified using the Montage PCR Cleanup Kit (Millipore). Cleaned PCR samples were quantified using a NanoDrop 1000 Spectrophotometer (Thermo Scientific). Sequencing was outsourced to Source Bioscience (Santa Fe Springs, CA, USA). The sequence was assembled and edited using Geneious Pro 8.1.7 (*Kearse et al., 2012*). Geneious was also used to extract the consensus sequence, which was 658 bp long and is deposited in GenBank (GenBank Voucher Number KU551261).

The electronic version of this article in Portable Document Format (PDF) will represent a published work according to the International Commission on Zoological Nomenclature (ICZN), and hence the new names contained in the electronic version are effectively published under that Code from the electronic edition alone. This published work and the nomenclatural acts it contains have been registered in ZooBank, the online registration system for the ICZN. The ZooBank LSIDs (Life Science Identifiers) can be resolved and the associated information viewed through any standard web browser by appending the LSID to the prefix http://zoobank.org/. The LSID for this publication is: urn:lsid:zoobank.org:pub:088D994A-9E1E-4324-A6DF-FCCC2B0E3437. The online version of this work is archived and available from the following digital repositories: PeerJ, PubMed Central and CLOCKSS.

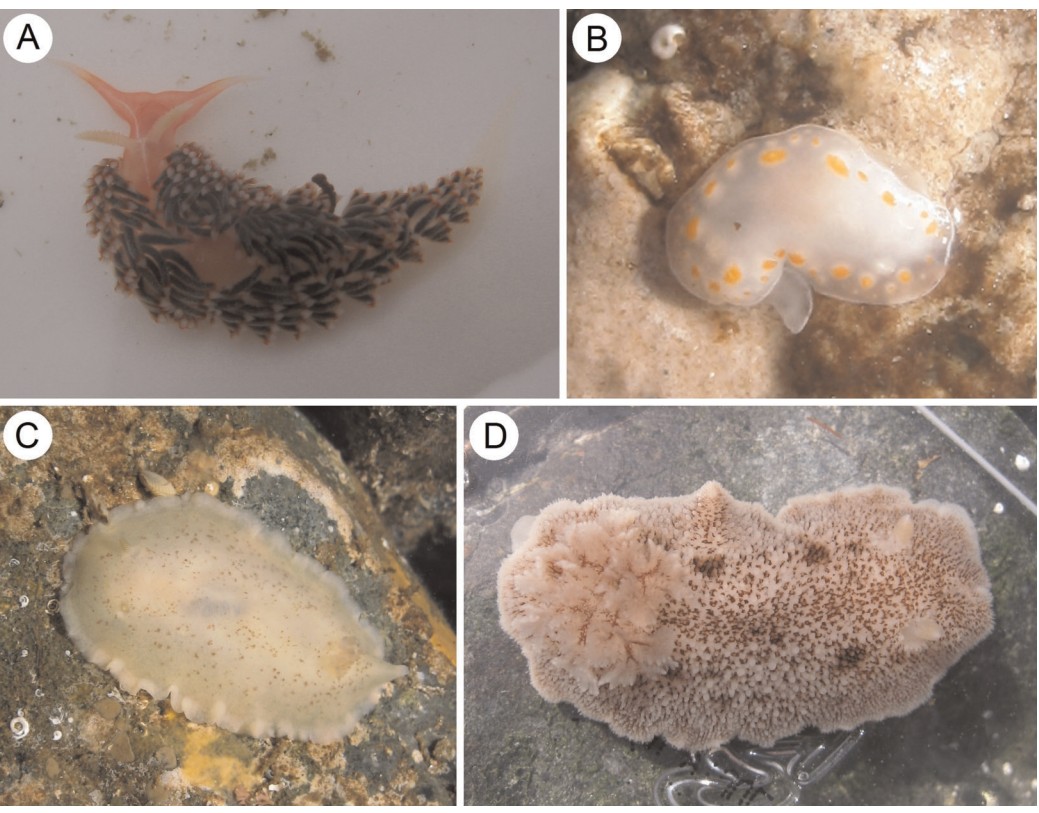

**Figure 1** **Species of heterobranch sea slugs found near Caldera, Atacama region, northern Chile (all specimens photographed in situ).** (A) *Phidiana lottini* (*Lesson, 1831*), Calderilla Beach, inside a valve of *Argopecten purpuratus* (Lamarck, 1819), L = 23 mm; (B) *Tyrinna delicata* (*Abraham, 1877*), Obispito Bay, L = 10 mm; (C) *Baptodoris peruviana* (d'Orbigny, 1837), Ramada Beach, L = 23 mm; (D) *Diaulula variolata* (d'Orbigny, 1837), El Pulpo Beach, L = 34 mm.

## RESULTS

### Systematics

### Heterobranchia

**Order Nudibranchia Cuvier, 1817**
**Superfamily Aeolidioidea Gray, 1827**
**Family Facelinidae Bergh, 1889**
**Genus *Phidiana* Gray, 1850**

**Type species** *Eolidia patagonica* d'Orbigny, 1836, by subsequent designation by Alder & Hancock (1855).

### *Phidiana lottini* (*Lesson, 1831*) (*Fig. 1A*)

*Eolidia lottini Lesson, 1831*: 290, pl. 14, fig. 6. *Cavolina lottini* d'Orbigny, 1837: 194. *Phidiana inca Gray, 1850*: 108; *Bergh, 1867*: 100, pl. 3, figs. 1–13; *Marcus, 1959*: 79, figs. 184–190; *Álamo & Valdivieso, 1997*: 85. *Phidiana lottini Schrödl, 1996a*: 41, pl. II,

fig. 13. pl. VII, fig. 41; *Schrödl, 2003*: 83, figs. 51, 63, 64, 88; *Schrödl, 2009*: 539; *Schrödl et al., 2005*: 7, pl. 2, fig. 17; *Uribe et al., 2013*: 52, fig. 3J; *Schrödl & Hooker, 2014*: 54, figs. 12, 13; *Uribe et al., 2014*: 167. A detailed chresonymy can be found in *Schrödl (2003)*.

**Material examined:** Two specimens collected in a tidal pool in rocky outcrops, Playa Brava (27°03′S; 70°49′W), Caldera, Región de Atacama, Chile (MZUC 39608); and one specimen collected inside an empty *Austromegabalanus psittacus* shell in Calderilla (27°05′S; 70°50′W), Caldera, Región de Atacama, Chile (MPCCL 90216A).

**Diagnosis:** Elongate body of silky white to sometimes reddish color, covered by 20–26 parallel rows of conspicuously colored cerata. Dorsum with a fine longitudinal white line. Cerata with bands of brown and orange at base and with bright whitish tips. Rhinophores annulate, yellowish white. Oral tentacles long and pinkish-white. Anterior foot corners slightly extended.

**Distribution:** *Phidiana lottini* has been recorded in Chile from Punta Blanca, Arica (18°29′S; 70°20′W) to the Guaitecas Islands (44 °S), southern Chile (*Schrödl, 2003*; *Schrödl & Hooker, 2014*). This species has also been recorded from Ancash, Isla Santa, Lima, and Callao (12°02′S), central Peru (*Uribe et al., 2013*; *Schrödl & Hooker, 2014*).

**Remarks:** *Phidiana lottini* is easily recognizable from other aeolid sea slugs found in northern Chile because of the cerata arranged in parallel rows and the presence of a white dorsal line between the rhinophores. This is a comparatively common nudibranch in the area, usually found in protected localities. Egg masses of this species are loosely coiled whitish spiral ribbons, of about 30 mm in diameter (see *Schrödl, 2003*).

## Superfamily Doridoidea

**Family Chromodorididae Bergh, 1891**

**Genus *Tyrinna Bergh, 1898***

**Type species** *Tyrinna nobilis Bergh, 1898* (= *Tyrinna delicata* (*Abraham, 1877*)), by monotypy.

### *Tyrinna delicata* (*Abraham, 1877*) (*Fig. 1B*)

*Doris delicata Abraham, 1877*: 211, pl. XXX, figs. 20–22. *Tyrinna nobilis Bergh, 1898*: 523, pl. 30, figs. 21–29, pl. 32, figs. 21–24; *Marcus, 1959*: 31, figs. 45–53; *Muniaín, Valdés & Ortea, 1996*: 265, figs. 2–6; *Schrödl, 1996a*: 22, pl. 3, fig. 15; *Schrödl, 1997*: 41; *Schrödl, 2003*: 31, figs. 15, 70; *Schrödl et al., 2005*: 4, pl. 1, fig. 8; *Schrödl & Millen, 2001*: 1146, figs. 1–6; *Schrödl, 2009*: 521; *Aldea, Césped & Rosenfeld, 2011*: 43, fig. 3C. *Uribe et al., 2013*: 48, fig. 2A. *Tyrinna pusae Marcus, 1959*: 33, figs. 54–64. A detailed chresonymy can be found in *Schrödl (2003)*.

**Material examined:** One specimen collected under rocks at low tide, in tidal pools in rocky outcrops, South of Obispito (26°45′51″S; 70°45′07″W), Caldera, Región de Atacama, Chile (MPCCL 90216B).

**Diagnosis:** Body oval-elongate, translucent-whitish, with opaque white lines surrounding the edges of foot and mantle. Dorsum smooth, with irregular and submarginal rows of orange spots, absent from the central region of mantle. Oral tentacles longitudinally enrolled.

Anterior part of foot bilabiate, forming a thick lip. Posterior end of the foot extending beyond the mantle in crawling individuals (see *Uribe et al. (2013)* for a more complete description).

**Distribution:** From Isla Blanca (09 °S), Ancash, Peru to Peninsula Valdés, in the Atlantic Magellan Strait (*Schrödl & Millen, 2001*; *Uribe et al., 2013*). This species has been also recorded in the Juan Fernández Islands, off central Chile.

**Remarks:** *Tyrinna delicata* is clearly distinguishable from other nudibranchs in northern Chile by the submarginal dorsal rows of orange spots, which are very visible in the translucent whitish mantle. This species, having a complex synonymy, was listed as *Tyrinna nobilis* until recent, however the discovery of the holotype of *Tyrinna delicata* (*Abraham, 1877*) by *Schrödl & Millen (2001)* gave priority to the latter name.

**Family Discodorididae Bergh, 1891**

**Genus *Baptodoris* Bergh, 1884**

**Type species** *Baptodoris cinnabarina* Bergh, 1884, by monotypy.

### *Baptodoris peruviana (d'Orbigny, 1837) (Fig. 1C)*

*Doris peruviana* d'Orbigny, 1837: 188, pl. XV, figs. 7–9. *Doriopsis peruviana Dall, 1909*: 203. *Platydoris punctatella Bergh, 1898*: 521, figs. 12–20; *Dall, 1909*: 203; *Schrödl, 1996a*: 23, pl. IV, fig. 27. *Dendrodoris peruviana Álamo & Valdivieso, 1997*: 85. *Platydoris peruviana Schrödl, 2003*: 34, figs. 17, 54, 71. *Baptodoris peruviana Fischer & Cervera, 2005a*: 515, figs. 1–8. *Uribe et al., 2013*: 51, fig. 3D. *Baptodoris*? *peruviana Schrödl & Hooker, 2014*: 48, fig. 4.

**Material examined:** One specimen collected under rocks at very low tide, Playa Ramada (27°00′S; 70°48′W) Caldera, Región de Atacama, Chile (MZUC 39607).

**Diagnosis:** Elevated, oval and slightly convex white-yellowish body, with minute brown spots over the notum which is densely covered by very small rounded caryophyllidia. Rhinophores and gills hyaline white, not elevated. Rhinophores are perfoliate with 7–10 lamellae. The branchial tuft consists of 6 uni-bipinnate gills, which form a circle around the anus at the posterior end of the body. Ventrally, the head is small with short digitiform oral tentacles. The foot is narrow, with the anterior edge notched at the mid-line and grooved. The notal margin is white and wider than the foot (see *Fischer & Cervera (2005a)* for a complete description).

**Distribution:** According to *Fischer & Cervera (2005a)*, this species has been recorded from South of San Lorenzo Island, Lima, Peru to Valparaiso, (33°02′S; 71°38′W) Chile.

**Genus *Diaulula* Bergh, 1884**

**Type species** *Doris sandiegensis* (Cooper, 1863), by monotypy.

### *Diaulula variolata (d'Orbigny, 1837) (Fig. 1D)*

*Doris variolata* d'Orbigny, 1837: 186, pl. 16, figs. 1–3. *Anisodoris marmorata Marcus, 1959*: 45, figs. 98–103; *Schrödl, 2003*: 41, figs. 21, 57, 75; *Fischer & Cervera, 2005b*: 174. *Uribe et al., 2013*: 48, fig. 2B. *Anisodoris marmorata Bergh, 1898*: 515, pl. 30, figs. 5–7 (non *Archidoris marmorata* Bergh, 1881); *Marcus, 1959*: 45, figs. 98–103. *Anisodoris rudberghi*

*Marcus & Marcus, 1967*: 69; *Schrödl, 1996a*: 25, pl. IV, figs. 21–22; *Schrödl, 1996b*. *Peltodoris marmorata Valdés & Muniaín, 2002*: 349, figs. 1D, 4, 5. A detailed chresonymy can be found in *Schrödl (2003*: 39*).

**Material examined:** One specimen collected under rocks at very low tide, North of Obispito (26°45′S; 70°45′W), 40 km N of Caldera, Región de Atacama, Chile (MZUC 39606).

**Diagnosis:** Whitish-yellowish body with minute black spots over the notum, which is densely covered by small and narrow caryophyllidia. Wide free mantle rim. Rhinophoral and branchial sheaths elevated, covered with caryophyllidia. Six to seven gills, ramified up to four-five times. Oral tentacles long and digitiform. Foot bilabiate, with upper lip notched. Lip cuticle smooth. Rhinophores have more than 15 lamellae (see *Schrödl (2003)* for a complete description).

**Distribution:** This species has been recorded in Chile from Arica (18 °S) to the Bahía de San Vicente (36 °S), and most recently from Ica, Perú (*Uribe et al., 2013*).

**Family Dorididae Rafinesque, 1815**

**Genus *Doris* Linnaeus, 1758**

Type species *Doris verrucosa* Linnaeus, 1758, by monotypy.

### *Doris fontainii d'Orbigny, 1837 (Fig. 2A)*

*Doris fontainii* d'Orbigny, 1837: 189, pl. 15, figs. 1–3. *Anisodoris fontaini Odhner, 1926*: 85, figs. 70–72, pl. 3, figs. 47–49; *Schrödl, 1996a*: 24, pl. III, fig. 19; *Schrödl, 2000b*: 73, fig. 2–3. *Doris fontainei Gay, 1854*: 76; *Valdés & Muniaín, 2002*: 346, figs. 1A–B, 2A–C, 3A–B; *Uribe et al., 2013*: 51, fig. 3E; *Schrödl & Hooker, 2014*: 47, fig. 2. *Archidoris fontaini Schrödl, 2003*: 45, figs. 24, 58, 76; *Schrödl, 2009*; *Schrödl et al., 2005*: 4, pl. 2, fig. 9; *Schrödl & Grau, 2006*: 5, fig. 2A–B.

**Material examined:** One specimen collected in a tidal pool at Playa El Jefe (27°03′46″S; 70°49′W), Caldera, Región de Atacama, Chile (MZUC 37642).

**Diagnosis:** Orange to brownish body coloration, with a highly arched and large body (up to 10 cm according to *Schrödl & Hooker (2014)*). Notum covered with many small (up to 5 mm in diameter) rounded tubercles. Five to seven tri- to quadripinnate gills. Gills and rhinophores surrounded by elevated sheaths with small tubercles. Oral tentacles triangular, grooved. Foot broad, anteriorly bilabiate and notched. Lip cuticle smooth (see *Schrödl (2003)* for a complete description).

**Distribution:** This species has been recorded from Ancash, Islote Ferrol, Peru (*Uribe et al., 2013*) to northern Argentina (*Valdés & Muniaín, 2002*).

**Remarks:** This species is easily recognizable due to its large size, brilliant orange body color and a mantle covered with conspicuous rounded tubercles. Of the examined specimens, none had the dark brown pigment between the tubercles, which *Schrödl et al. (2005)*, regarded as characteristic of central and northern Chilean specimens. This was the most common species in the area; they are usually found in the subtidal zone but specimens were also collected from tidal pools at low tide. According to some commercial divers this species is common below 3 m depth near Bahía Inglesa (27°07′S; 70°52′W), south of Caldera.

 

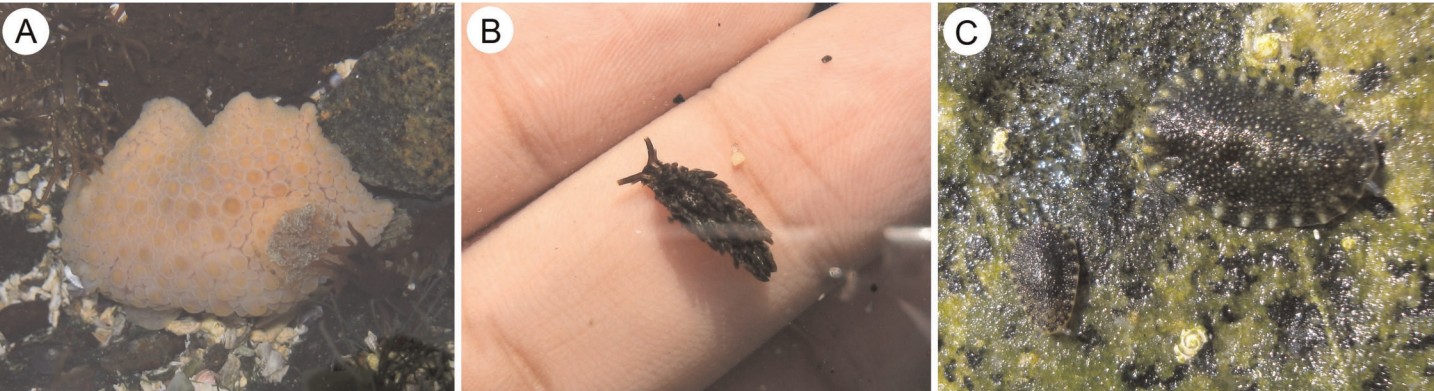

**Figure 2** **Species of heterobranch sea slugs found near Caldera, Atacama region, northern Chile (all specimens photographed in situ).** (A) *Doris fontainii* d'Orbigny, 1837, Playa El Jefe, L = 54 mm; (B) *Aplysiopsis* cf. *brattstroemi* (*Marcus, 1959*), Brava Beach, specimen found among filamentous algae in tidal pool, L about 4 mm; (C) *Onchidella marginata* (Couthouy in *Gould, 1852*), Playa El Pulpo, L = 12 mm (largest specimen).

**Order Pleurobranchomorpha Schmekel, 1985**
**Superfamily Pleurobranchoidea Gray, 1827**
**Family Pleurobranchidae Gray, 1827**
**Genus *Berthella* Blainville, 1824**

**Type species** *Bulla plumula* Montagu, 1803, by original designation.

### *Berthella schroedli* sp. nov.

urn:lsid:zoobank.org:act:9F1D698F-96FB-40B0-A972-3C1F6F15014C (Figs. 3A–3C, 4A–4D, 5A, 5B and 6C).

**Type material:** Holotype MPCCL 90216C, paratypes: LACM 3327 (4 specimens), MPCCL 90216D (4 specimens); other material: CPIC 000827 (5 specimens). All the type material collected at the type locality and preserved in ethanol 96%.

**Type locality:** Playa El Pulpo (27°01′22″S; 70°48′30″W), Comuna de Caldera, Región de Atacama, Chile, intertidal under sunken rocks in rocky coast, 1 m depth, 29 December 2012, coll. & leg. JF Araya.

**Diagnosis:** Intertidal *Berthella* species with a dark brown-reddish shell decorated with pale radial lines; visible through the translucent yellowish mantle, with an oval and slightly crenulated margin and very small tubercles covering the notum.

**Description:** Body reaching lengths up to 25 mm in fully extended living specimens (Figs. 3A, 3B and 6C). Body uniformly pale yellowish, translucent; with an internal shell of brownish-reddish color, visible through the mantle. Mantle with a smooth appearance, but with very small tubercles covering the dorsum which gives the animal, at high magnification, a somewhat wrinkled appearance. The mantle processes do not show obvious spicules. Dark and minute eyes located behind the base of the rhinophores, hidden beneath the anterior edge of the mantle (Fig. 3B). Notum wide, oval and slightly crenulated, with a broad free margin around. Gill and foot covered by the notum in living specimens, and oral veil and rhinophores partially covered in their posterior part.

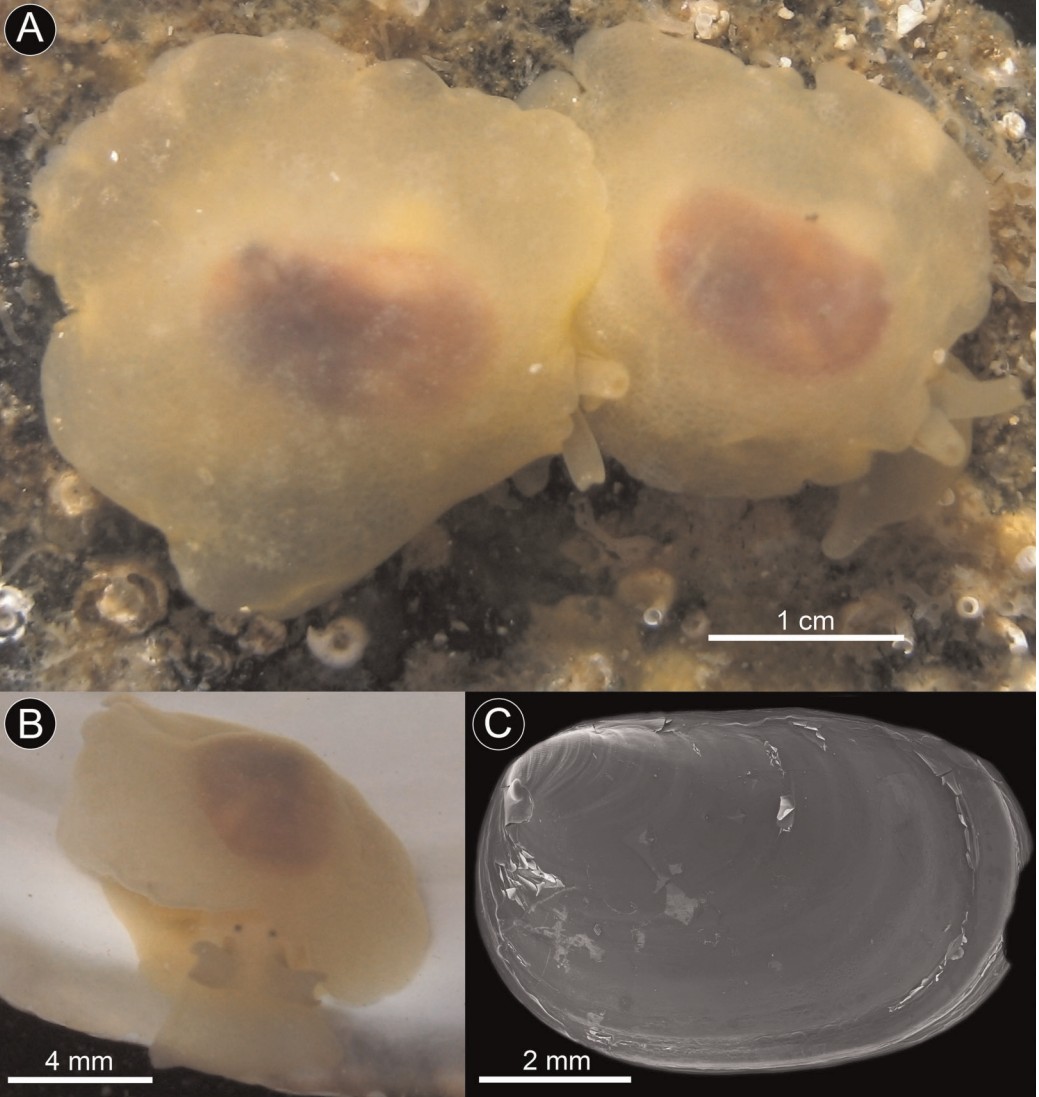

**Figure 3 *Berthella schroedeli* sp. nov.** (A) Specimens photographed in situ, under rocks at low tide, Aguas Verdes; (B) Detail of specimen showing the eyes; (C) SEM image of shell (LACM 3327).

Mantle lacking an anterior notch. Rhinophores short and stout, joined together at the base. Foot bilabiate anteriorly. Oral veil trapezoidal, protruding from the mantle. Gill located on the right side of the body, lying longitudinally between the mantle and the foot; it is attached to the body for more than half of its length. Gill bipinnate, with 13 pinnae on either side of the rachis. Rachis smooth, lacking tubercles. Anus located dorsal to the central area of the gill. Egg masses are small white spiral ribbons, up to about 25 mm in diameter (Fig. 6C).

*Shell*: Shell fully internal, flattened, rectangular/oval in shape, elongate and located centrally in the dorsal area, where it covers completely the viscera. Shell reddish brown in color, somewhat nacreous/iridescent, with radial rays of pale yellowish which are visible through the mantle in living specimens. Margins of shell sharp and fragile. Protoconch of about 300 μm in diameter, smooth under low magnification. Teleoconch with fine

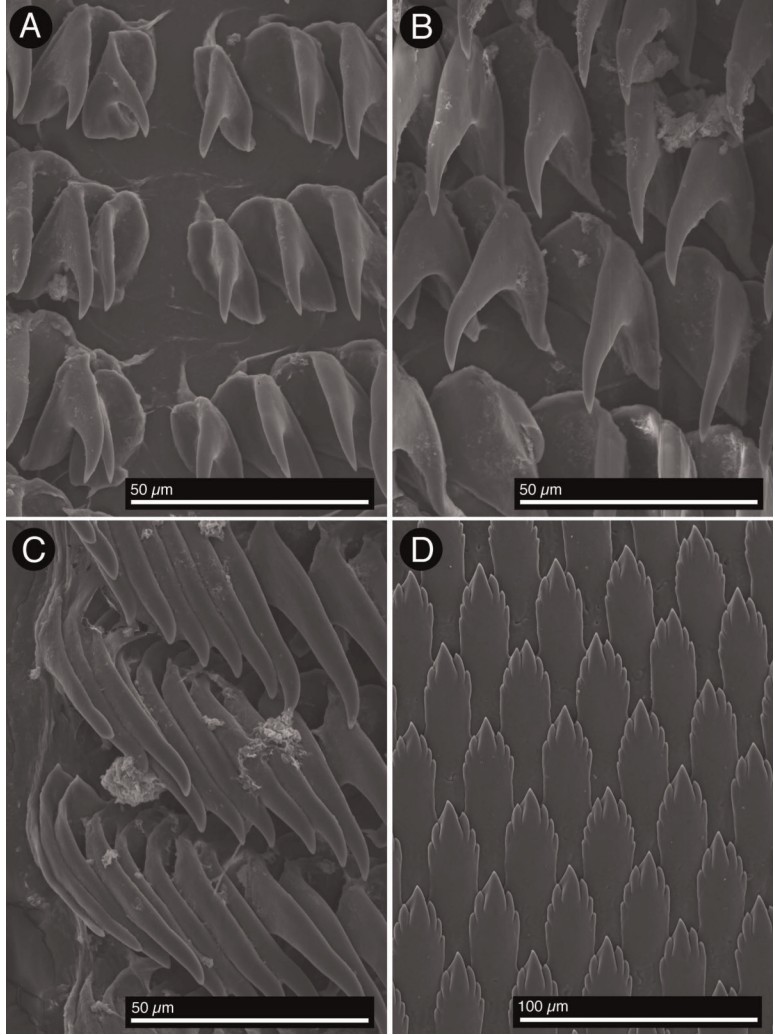

**Figure 4** *Berthella schroedeli* **sp. nov., SEM images (LACM 3327).** (A) Radular teeth, central portion of the radula; (B) Outermost radular teeth; (C) Lateral teeth, middle portion of the half row; (D) Detail of the jaw platelets.

concentric ridges crossed by very fine radial striae, the first whorls have a cancellated sculpture (Fig. 3C). *Radula*: Radular formula: 50–53 × 45–56.0.45–56. Radular teeth hook-shaped lacking denticles (Fig. 4A). Innermost lateral teeth slightly smaller than those from the middle portion of the half row (Fig. 4B). Outermost lateral teeth with a much more elongate cusp than the mid laterals (Fig. 4C). Jaws with elongate cruciform elements rather slender, elongate and lanceolate with a narrower base; each element consisting of a central cusp flanked by 2–3 denticles on either side of a prominent central cusp (Fig. 4D). *Reproductive system*: The ampulla is long and muscular, merging proximally into the female gland complex. The penis is wide, with an elongate tip; it connects proximally into a short deferent duct that splits into the prostate and the elongate, muscular penial gland. The prostate is convoluted and connects proximally to the female gland complex. A small, unidentified glandular structure connects distally

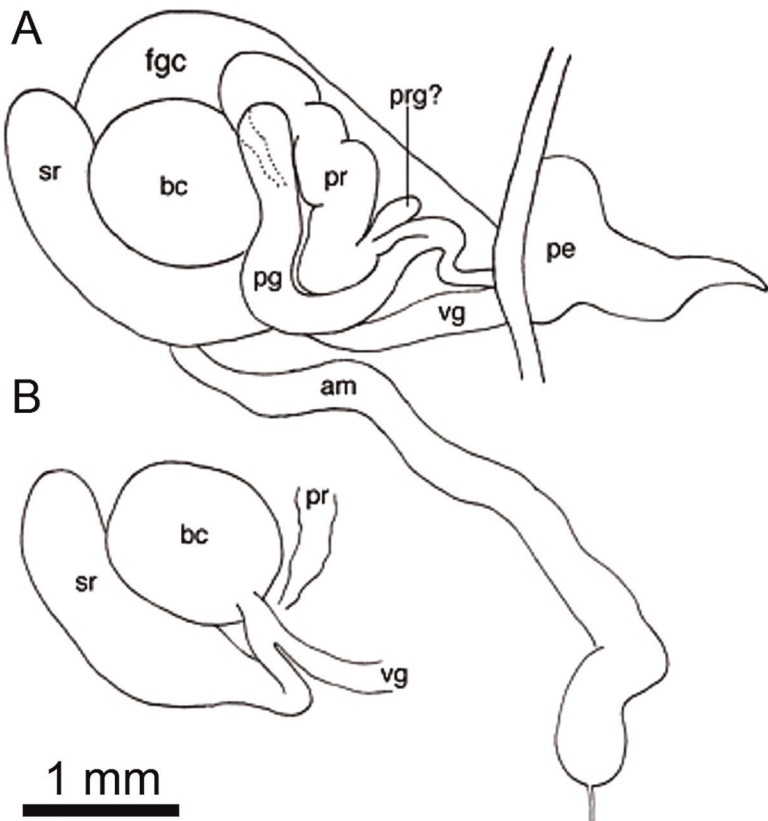

**Figure 5 Reproductive anatomy of *Berthella schroedli* sp. nov.** (A) Dorsal view of the reproductive system; (B) Detail of some organs covered by the prostate and penial gland. Abbreviations used are: am, ampulla; bc, bursa copulatrix; fgc, female gland complex; pe, penis; pg, penial gland; pr, prostate; sr, seminal receptacle; vg, vagina.

into the prostate and is here referred to provisionally as prostatic gland (prg? in Fig. 5A). The vagina is elongate, straight; it narrows and connects to the round and large bursa copulatrix. The seminal receptacle is elongate, muscular and about twice as long as the bursa copulatrix; it connects to the vagina before it enters the bursa copulatrix. A uterine duct could not be observed (Fig. 5).

**Habitat:** This species is found exclusively under rocks sunken at low tide in an almost infaunal habitat; it can be found associated to encrusting sponges, bryozoans, encrusting algae and to communities of micromollusks including *Acar pusilla* (*Sowerby, 1833*), *Brachidontes granulata* (Hanley, 1843), *Liotia cancellata* Gray, 1848 and *Mitrella unifasciata* (*Sowerby, 1832*).

**Distribution:** This species is somewhat rare but broadly distributed in the area of study; small populations were found only in four localities, in about 40 km of coast, always under rocks. According to *Schrödl (2003)* this genus has records in southern, South America from the southernmost Patagonian shelf (Burdwood Bank), southeastern Atlantic Ocean to southern Chile and north to Quiriquina Island, central Chile. The genus thus extends its distribution in Chile more than 1,100 km to the north.

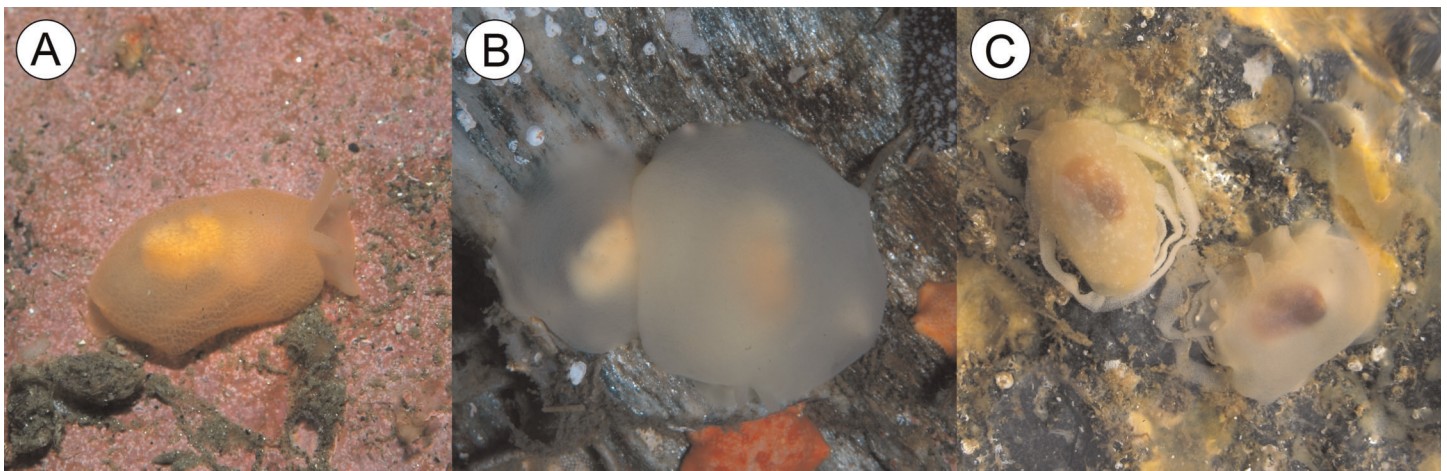

**Figure 6 Chilean *Berthella* species.** (A) and (B) specimens of *Berthella platei* (*Bergh, 1898*) photographed in situ, Caleta de Arena, 20 m depth and Valdivia respectively (photos (A) and (B) courtesy of Dirk Schories); (C) *Berthella schroedli* sp. nov., specimen sitting on egg masses, Obispito, Caldera.

**Etymology:** Named in honor of Michael Schrödl (Zoologische Staatssammlung München, Munich, Germany), for his extensive contributions to the Chilean opisthobranchs.

**Remarks:** Of the 16 valid species of *Berthella* known worldwide (*Hermosillo & Valdés, 2008*), only two have been reported for southern South America: *Berthella patagonica* (d'Orbigny, 1837) and *Berthella platei* (*Bergh, 1898*). The western Atlantic *Berthella patagonica*, distributed from Central Argentina to Peninsula Valdés, southern Argentina (*Schrödl, 2003*), differs from the new species in having smaller body dimensions, with a very narrow free mantle rim and a notum apparently lacking a porous texture and not covering completely the foot which, in contrast to the new species, has a quadrangular outline (*Schrödl, 1999*; *Schrödl, 2003*). The Magellanic *Berthella platei*, distributed from the Burdwood Bank, southeastern Atlantic Ocean to Quiriquina Island, Central Chile (*Schrödl, 1999*), differs from the new species in having a more translucent body, of uniform pale pink to pale orange or whitish coloration of living animals (Figs. 6A and 6B), a higher number (15–24) of branchial lamellae versus 11–14 in *B. schroedli* sp. n. and a paler internal shell, translucent brown to greyish in color, in contrast to the characteristic reddish-brown shell with faint whitish axial streaks of the new species. The radular formula and the elements of the jaws also differ; *Berthella schroedli* sp. n. have fewer radular rows and less teeth per half row than *B. platei*, and it has also larger elongate and lanceolate elements with a narrower base and thin denticles, while *B. platei* have smaller and more triangular elements with a broader base and slightly broader denticles (see *Schrödl, 1999*). The shell length in relation to the body size in *B. schroedli* is also comparatively larger than in *B. platei*. In regard to their habitat; the new species has been found almost solely under sunken rocks in relatively shallow water in the intertidal; while *Berthella platei* is found only subtidally, living in the ocean floor usually under 5 m depth (Dirk Schories, 2013, personal communication). A BLAST-n of the COI sequence of *B. schroedli* sp. n. returned that the most similar sequence belongs to *Berthella*

*plumula* (AY345025) and is only 84% identical. The sequence of *B. schroedli* sp. n. is only 83% identical to a sequence of *Berthella platei* (FJ917492), providing additional evidence that this species is distinct.

Other Eastern Pacific species of *Berthella* include *Berthella agassizi* (*MacFarland, 1909*); *Berthella californica* (Dall, 1900); *Berthella grovesi Hermosillo & Valdés, 2008*; *Berthella martensi* (*Pilsbry, 1896*); *Berthella stellata* (*Risso, 1826*) and *Berthella strongi* (*MacFarland, 1966*). All these species differ from *Berthella schroedli* sp. n. in their subtidal rather than intertidal habitat, and also chiefly in their external coloration, by having opaque white spots (*B. agassizii, B. strongi*) or light brown spots and/or an orange body with dark brown lines and spots (*B. martensi*), a marginal notal band (*B. californica*), dark spots in the middle of thick opaque white ringlets (*B. grovesi*) or a dorsal streak of white running perpendicularly across the notum, which is translucent white or honey colored (*B. stellata*).

**Order Sacoglossa Ihering, 1876**

**Superfamily Limapontioidea Gray, 1847**

**Family Hermaeidae Adams & Adams, 1854**

**Genus *Aplysiopsis* Deshayes, 1853**

**Type species** *Aplysiopsis elegans* Deshayes, 1853, by monotypy.

### *Aplysiopsis cf. brattstroemi (Marcus, 1959) (Fig. 2B)*

*Hermaeina brattstroömi Marcus, 1959*: 21, figs. 21–27. *Aplysiopsis brattstroemi Schrödl, 1996a*: 45, pl. VIII, fig. 52; *Fischer & Cervera, 2005a*: 167; *Jensen, 2007*: 279.

**Material examined:** One specimen photographed alive (not collected); on filamentous algae in tidal pool at very low tide, Playa Brava (27°03′S; 70°49′W), Caldera, Región de Atacama, Chile.

**Diagnosis:** Body minute, up to about 5 mm in examined specimen, with an elongated body, narrowed anteriorly; of brown to deep greenish-black color, with two clear areas at the sides of the head. Several rows of flat longitudinal cerata in the border of the mantle. Enrolled rhinophores. Size up to about 3 cm (see *Marcus (1959)* for a complete description).

**Distribution:** *Aplysiopsis brattströmi* has a discontinuous distribution from Antofagasta (23°39′S; 70°25′W), to Bahía de Coliumo (36°32′S; 72°57′W) in Chile (*Schrödl, 1996a*). The definite allocation of this specimen is currently not possible as, unfortunately, it was not collected.

**Order Systellommatophora Pilsbry, 1948**

**Superfamily Onchidioidea Rafinesque, 1815**

**Family Onchidiidae Rafinesque, 1815**

**Genus *Onchidella Gray, 1850***

**Type species** *Onchidium nigricans* Quoy & Gaimard, 1832, by subsequent designation by Fischer and Crosse (1878).

### *Onchidella marginata (Couthouy in Gould, 1852) (Fig. 2C)*

*Peronia marginata* Couthouy in *Gould, 1852*: 292; Atlas, 1856: pl. 22, figs. 386a–c. *Onchidium chilense Gay, 1854*: 120. *Onchidella marginata Marcus, 1959*: 16, figs. 17–20. *Dayrat, 2009*: 13. *Rosenfeld & Aldea, 2010*: 35, figs. 1A–B. A more complete synonymy can be found in *Dayrat (2009)*.

**Material examined:** Ten specimens collected under small rock slabs at low tide, Playa El Pulpo (27°03′S; 70°49′W), Caldera, Región de Atacama, Chile (MZUC 280316).

**Diagnosis:** Body elongate ovate, narrowed anteriorly; back very convex, deep greenish-black, very thickly covered with minute tubercles; margin ornamented with alternate bars of black and white; head broad, bilobed in font, and projecting considerably beyond the mantle when the animal is in motion, of a pale yellow color, tinted bluish about the mouth; tentacles rather long, and bulbous at the extremity, pale slate-color, except at the tips, which are back; under side of the mantle pale yellowish, becoming greenish at the margin, where it shows alternate bands of green and pale yellow (see *Gould (1852)* for a complete description).

**Distribution:** *Onchidella marginata* has a discontinuous distribution from Iquique (20 °S) to the Magallanes Strait (55 °S) in Chile, and to the Isla de los Estados in the South Atlantic of Argentina (*Rosenfeld & Aldea, 2010*).

**Remarks:** This is the only pulmonate sea slug found in Chile (*Valdovinos, 1999*; *Dayrat, 2009*); it is usually found in small communities living under rocks and camouflaging against their surroundings. In the area under study this species share its habitat with other mollusks as the limpet *Lottia orbignyi* (*Dall, 1909*), and the chitons *Chaetopleura peruviana* (Lamarck, 1819) and *Radsia barnesi* (Gray, 1828).

## DISCUSSION

The present work updates the knowledge on the scarcely known marine fauna of northern Chile (in particular from the Región de Atacama); from the 65 species of sea slugs (only including Nudibranchia and Pleurobranchoidea) recorded to live in Chilean waters (*Schrödl, 2003*), eight species were recorded in the Región de Atacama, accounting for about 12% of the Chilean sea slug fauna. All of the species occurring in the area have widespread ranges in the southeastern Pacific Ocean, from Ancash, Peru to the Strait of Magellan, in southern Chile and in the South Atlantic Ocean, to Peninsula Valdés, in Argentina (Table 1). With the exception of *Berthella schroedli* sp. n., all of the species found in the Región de Atacama also occur in central and southern Chile. The absence of species previously cited for the area (*Schrödl, 1996a*; *Schrödl, 2003*; *Schrödl & Hooker, 2014*), for example *Corambe lucea Marcus, 1959*; *Janolus rebeccae Schrödl, 1996a*; *Schrödl, 1996b*; *Okenia luna Millen et al., 1994* and *Thecacera darwini* Pruvot-Fol, 1950, among others, could be explained due to the limit of sampling depth, which was restricted to the lower intertidal areas with a maximum of 2 m depth.

Heterobranch sea slugs have been rarely treated in studies reviewing the biodiversity of mollusks from northern Chile (e.g. *Marincovich, 1973*; *Guzmán, Saá & Ortlieb, 1998*), despite the comparatively high number of species recorded in the country. This is in part explained by the current lack of experts working actively in the field and the

difficulties involved in collecting and preserving marine slugs. The finding of a new species of *Berthella* in northern Chile also highlights the need of further studies in the area or in northern Chile in general, which have recently revealed new invertebrate species (*Reiswig & Araya, 2014*; *Collado, 2015*) or new distributions for obscure or rare species, both from shallow and deeper waters (e.g. *Araya & Aliaga, 2015*; *Araya & Araya, 2015b*; *Araya, Aliaga & Araya, 2015*; *Araya, 2015c*; *Fischer, van der Velde & Roubos, 2007*; *Labrín, Guzmán & Sielfeld, 2015*).

## ACKNOWLEDGEMENTS

We are very grateful to Marta Araya (Caldera, Chile) for her assistance in field collecting, to Carlo Magenta Cunha (Academy of Natural Sciences of Drexel University, Philadelphia, USA), and to Cecilia Osorio (Universidad de Chile, Santiago, Chile) for their help with essential bibliography, to Dirk Schories (University of Rostock, Rostock, Germany) for his help with the images and information on *Berthella platei* from southern Chile and to Michael Schrödl (Zoologische Staatssammlung München, Germany) and two anonymous reviewers for their helpful corrections and suggestions on the manuscript.

### Funding

The authors received no funding for this work.

### Competing Interests

The authors declare that they have no competing interests.

### Author Contributions

- Juan Francisco Araya conceived and designed the experiments, performed the experiments, analyzed the data, contributed reagents/materials/analysis tools, wrote the paper, prepared figures and/or tables, reviewed drafts of the paper.
- Ángel Valdés conceived and designed the experiments, performed the experiments, analyzed the data, contributed reagents/materials/analysis tools, wrote the paper, prepared figures and/or tables, reviewed drafts of the paper.

### Data Deposition

The research in this article did not generate nor collect any raw data/code.

### New Species Registration

The following information was supplied regarding the registration of a newly described species:

Publication LSID: urn:lsid:zoobank.org:pub:088D994A-9E1E-4324-A6DF-FCCC2B0E3437.

*Berthella schroedli* sp. nov. LSID: urn:lsid:zoobank.org:act:9F1D698F-96FB-40B0-A972-3C1F6F15014C.

## Supplemental Information

Supplemental information for this article can be found online at http://dx.doi.org/
10.7717/peerj.1963#supplemental-information.

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
