# Peer review of "Shallow water heterobranch sea slugs (Gastropoda: Heterobranchia) from the Región de Atacama, northern Chile"

_PeerJ, doi:10.7717/peerj.1963_

## Round 0.1 · original submission · Major Revisions

I now have comments back from 3 referees who have each offered extensive comments on your submission. While each is supportive of the work, they are all consistent in their request for substantial revision prior to publication. They question why photos are not provided for every species in the survey, and one taxonomic expert questions some of the identifications, asking for dissections or microscopic morphological information added to the text for confirmation of the IDs. Even the most positive of the referees comments that the Remarks section of most species is very poor and there are mistakes in some species synonymic lists that must be corrected prior to publication. Finally all three request careful review for English grammar and usage throughout, and one has provided a marked up PDF with suggestions for you in this regard. Not being a taxonomic expert in these organisms myself, I have to concur with the opinions of the specialists who provided the review and ask that you clearly address their comments and concerns in your revision. Overall, however, I see nothing in the referee comments to indicate that the manuscript could not be revised to become suitable for publication. Because the requested revisions and taxonomic confirmation are substantial, I anticipate sending it back to these referees for re-evaluation after you complete the revisions.

·

Basic reporting

No comments; authors are trying to include museum numbers

Experimental design

Report on „Heterobranch sea slugs...“ by Araya & Valdés

This is a faunal study adding valuable information to the generally poor knowledge of marine invertebrates in northern Chile. Several sea slug species are externally illustrated and a new species of Berthella is described externally and internally. The anatomical description seems sound and the intertidal habitat of this new species is interesting. It would be good to include sequences to support or reject morphology-based conclusions on the novelty of B. schroedli; perhaps this could be done in a revised version.


My main problems are:
- The specimen illustrated as D. punctuolata (Fig. 1D) looks like a perfect P/D. variolata to me. I found many of them along the central and northern Chilean coast. Compare Schrödl 2003: fig 21A for colour variation of specimens from a single population…

- The “P. marmorata” should be checked for small caryophyllidia in between larger tubercles, please; if present, the specimen shown / species should be treated as synonym of D. variolata (also see below)

- The “Ercolania evelinae” in Fig. 1F does not look like those described by Marcus (1959) or Schrödl (1996), which have a single marginal row of just up to 12 cerata. Also, rhinophores appear to be somehow bifid or enrolled (?) while Ercolania evelinae has digitiform rhinophores. As a rough guess, that specimen looks more similar to Aplysiopsis brattstroemi (see Schrödl 1996).

All these issues could be clarified easily by light microscopy / SEM (tubercles) or by dissecting specimens and comparing anatomy with some relevant literature, such as Marcus (1959) and Schrödl (2003). A revised version should really address these major points and consider consequences, e.g. regarding synonymy lists, distributional ranges and information presented in Table 1.

Further comments and suggestions:

P 3, l 49: please use T. delicata rather than T. nobilis; also in legend of Fig. 1

p. 5: Check spelling of Eolidea patagonica d’Orbigny, 1836

add museum numbers throughout

p 6, l line 142: not the northernmost record from Chile; check Schrödl’s (2003) record from Arica.

P 7, l 172-173: actually it’s not a dorsal row but there are 2-3 submarginal dorsal rows

P 9, l 220-222: also occurring off northern Argentina; some records from northern Chile and Peru regarded as dubious by Schrödl (2003)

P 9, l 243: The southern specimens identified as D. variolata by Aldea et al. were doubted / disregarded to belong to that species by Uribe et al 2013 and Schrödl & Hooker 2014; Check the latter study for current reliable geographic ranges

P 10, l 248: all exemplars assigned to this species that were living or preserved adequately (rather fresh material in ethanol rather than formalin) clearly had at least some smaller caryophyllidia, i.e. larger tubercles look similar to those of Peltodoris but there are many slender caryophyllidia in between, as mentioned by Marcus (1959) and Schrödl (2003); I am sure that Bergh’s species should be better called Diaulula variolata, as emphasized by Schrödl & Hooker (2014), but eventually this decision is up to the authors.

P 11, l 300: eyes are at or under the base of rhinophores?

P 13, l 341: should read Munich

P 13, l 366: do they also differ regarding subtidal versus intertidal habitat?

Check text / legends for “schroedeli”

Validity of the findings

see above

Additional comments

All the best, Michael

Reviewer 2 ·

Basic reporting

In general, the manuscript needs a deep revision of the language and the style. There are many and important mistakes. The table also needs revision and should be cited in the introduction and not only in the distribution of Berthella. The table is missing one species, dots, brackets, coordinates…
Figures are good. But letters A and B in Figure 4 are very small in comparison with the figure, scale, etc. Also, there is something like a gland close to the prostate that is not labeled and no explained in the text. What is it? Figure 4A and 4B should be in the text. 4B is missing in the text and 4A (in the text) does not correspond with the figure.

Experimental design

They are ok but, in the notes from the author, he says that they are sequencing the new Berthella. If they pretend to include that information, they should have mentioned it before. That would change the manuscript.

Validity of the findings

In general, the information presented in the manuscript is very poor. Remarks of the species are very short and uninteresting. Descriptions seem not to be originals.

Additional comments

In general, the manuscript needs a deep revision of the language and the style. There are many and important mistakes. The table also needs revision and should be cited in the introduction and not only in the distribution of Berthella. The table is also missing Ercolania evelinae, dots, brackets, coordinates…
Figures are good. But letters A and B in Figure 4 are very small in comparison with the figure, scale, etc. Also, there is something like a gland close to the prostate that is not labeled and no explained in the text. What is it? Figure 4A and 4B should be in the text. 4B is missing and 4A does not correspond with the figure.
In general, the information presented in the manuscript is very poor. Remarks of the species are very short and uninteresting.

Other comments:
- What is “Chile and Programa de Doctorado en Sistemática y Biodiversidad” doing in the address of the first author?
- The authors talk about “the coasts” throughout the text. It should read “the coast” since it is only one.
- “mollusks” or “molluscs”?. Inconsistent.
- “contribute additional information”? I does´t make sense.
Please, carefully review the English throughout the text.

Systematics:
Phidiana:
- Which specimen refers MPCCL XXXX to? Not clear.
- Commas after authors (throughout the text).
- The description of Phidiana lottini is too general. Please, give more details. For example, the cerata are white only in their tips, etc.
- The Remarks of Phidiana lottini are not good. There are other species of that genus with those features. It needs a better discussion.
Tyrinna:
- commas in references after authors.
- If the dorsum is smooth, it is clear that it does not have tubercles.
- I don´t understand what the authors mean by “(After Uribe et al. 2013)” and the same in the remaining descriptions. Besides, in Uribe et al. (2013), this species is not mentioned, so the distribution is not correct.
- Remarks very poor.
Baptodoris:
- Again, what do you mean by (“After Fisher & Cervera, 2005a”?)
- Remarks: very poor.
Diaulula:
- The description of the specimen does not really match the picture. Small caryophyllidia? They seem very big to me. And what about the brown coloration? Where is the yellow?
- Again…why “After, Schrödl, 2003”?
- Distribution: and the Falkland Islands, too.
- Remarks: very poor.
Peltodoris, Ercolania, Onchidella:
- Same general comments as the species before.
- Fig. 1E should not be in bold.
- Schrodl or Schrödl?
- Picture of Onchidella marginata?
- Reference for the distribution of Onchidella marginata?
Berthella schroedli sp. nov.
- 18x7 mm???
- The information in “Type material” needs to be explained. What these refers to?? MZUC XXXX. Paratypes 1-3 LACM XXX-XXX, paratypes 4-6 MPCCL XXX-XXX???
- The diagnosis is not very complete.
- In line 21, the Fig 4A does not make any sense.
- The SEM picture of the jaws (Fig.3D) should be before the ones of the radula. It should be referred as Fig.3A. SEMs for the radula should be 3B, C and D. Change in the plate.
- The radular formula 50 x 45.0.45 is for all the specimens? How many specimens have the author dissected? This should be mentioned in the text.
- Reproductive system: In the text, Fig.4B needs to be mentioned. What is that “gland” close to the prostate? Please, explain it.

References: Many references are missing in the reference section (Broderip & Sowerby, 1832; Sowerby, 1832, 1833; d’Orbigny, 1835-1847; Hupé in Gay, 1854; Araya, 2015a? b? among many others), and also in the text. The format of the reference section also needs to be reviewed: missing brackets, dots, …

Reviewer 3 ·

Basic reporting

The manuscript is well written but English needs a review in few sections.

Experimental design

The research presented is original. I have no concerns about this section.

Validity of the findings

The most relevant topic of the manuscript is the description of a new species, which, based in the data presented seems to be really undescribed. Other species were already recorded in the region or nearby areas, and are not very relevant.

Additional comments

I have included comments in the attached pdf file. These include corrections and questions. Why the manuscript does not include photographs of all species recorded?
The Remarks section of most species is very poor and there are mistakes in some species synonymic lists. But overall, the manuscript is well written and the data relevant.

Annotated reviews are not available for download in order to protect the identity of reviewers who chose to remain anonymous.

---

## Round 0.2 · Minor Revisions

I am sorry for the delay in getting this back to you - we were waiting to hear back from the second referee who was out of touch until now. Both referees are satisfied with your revisions and feel that the revised manuscript is now acceptable for publication pending some minor corrections to the text. The only reason I have returned the manuscript with minor revisions is so that you can consider the comments of the referee who provided some additional information regarding the potential synonymy of marmorata with the senior variolata. Although I am willing to accept the manuscript regardless, I suspect that you will likely want to add this information to the manuscript prior to publication, so I am returning it to you so that you have the option of adding this information before the paper goes to press. I should be able to move it along into production quickly when you return a revision with these minor corrections.

·

Basic reporting

The authors greatly improved their manuscript and followed most of my species ID suggestions. There is still a problem regarding "Peltodoris marmorata", see below

Minor details:
page 2, line 17: omit "of the Región de Atacama, in" since its repeated in the next sentence

Page 3, line 13: Bergh (1898) is missing here

Page 4, line 16: omit "either" (?)

Page 5, line 21: omit "Infraclass" and put "Opisthobranchia" into quotation marks

Page 6, line 24: for details on egg masses you may add "(see Schrödl, 2003)"

Page 14, line 27: single specimen, so it should read ".. it was ..."

page 17, 2: should read "Munich" (or omit since "München" is already mentioned)

Experimental design

no comments

Validity of the findings

It's good to provide a COI barcode of the new Berthella. But why not mentioning the Genbank number in the species description? Please provide at least some analysis or comparison with B. platei.


I've seen many Diaulula variolata-like dorids in (central and northern, never southern!) Chile. Whitish to mottled or dark specimens: All recently and ethanol preserved (!) specimens I examined more closely finally showed at least some caryophyllidia. So they are NOT Peltodoris marmorata. Please, check your synonymy! Page 9, line 27: the Schrödl 2003 specimens and figs belong to Diaulula variolata, since they have caryophyllidia.

Considering "Peltodoris marmorata" a distinct species from Diaulula variolata you could reason that Bergh (1898) did not mention caryophyllidia, that neither Marcus (1959) nor Valdés & Muniaín (2002) found caryophyllidia, and that I just by chance only found many many Diaulula but never any Peltodoris in Chile. However, I would think it is more parsimonious to suspect that Bergh (1898) and Marcus and later authors overlooked the small (!) caryophyllidia between the big (!) normal tubercles; old or formalin preserved specimens could have easily lost their spicules... And I noticed that your own, single specimen of "Peltodoris" is really small! Perhaps you'll have a closer look via SEM? No apical sensory knob on the slender and small "tubercles"? Even if not, maybe caryophyllidia develop later, between the big tubercles?

Well, I just had the chance to recheck the Bergh material of variolata and the marmorata types: There are some more or less visible caryophyllidia (old material from 1898!) in variolata and the paralectotype of marmorata. The lectotype, unfortunately, is quite squeezed and the back in bad condition. But even there: some slender tubercles seem to show the typical apical knob of caryophyllidia and there are remainders of strong peripheral spicules. In my opinion, there is very little reason to doubt the synonymy of marmorata with the senior variolata. You can cite this new information as personal communication, if you want.

The Aldea et al 2011 specimens are neither D. variolata nor P. marmorata but almost certainly Diaulula punctuolata (as already stated in Uribe et al 2013). Please adjust and check the synonymy carefully

Additional comments

Sorry for emphasizing these variolata / marmorata points again. They are important to avoid later taxonomic confusion. The history of Chilean dorids was / is already complicated enough...

I'm ticking "minor revisions" because I trust you'll seriously consider the taxonomic problems emphasized above.
If in doubt, please feel free to send me for rerereview.

Reviewer 2 ·

Basic reporting

No comments. All previous concerns have been addressed.

Experimental design

No comments. All previous concerns have been addressed.

Validity of the findings

No comments. All previous concerns have been addressed.

---

## Round 0.3 · accepted · Accept

I have now reviewed your changes and I believe that your responses to the referee comments are satisfactory to all, so I am happy to accept your manuscript and move your paper into final production.